# Framework for Testing Robustness of Machine Learning-Based Classifiers

**DOI:** 10.3390/jpm12081314

**Published:** 2022-08-14

**Authors:** Joshua Chuah, Uwe Kruger, Ge Wang, Pingkun Yan, Juergen Hahn

**Affiliations:** 1Department of Biomedical Engineering, Rensselaer Polytechnic Institute, Troy, NY 12180, USA; 2Center for Biotechnology and Interdisciplinary Studies, Rensselaer Polytechnic Institute, Troy, NY 12180, USA

**Keywords:** classification, biomarker, machine learning, algorithms, artificial intelligence, omics analysis

## Abstract

There has been a rapid increase in the number of artificial intelligence (AI)/machine learning (ML)-based biomarker diagnostic classifiers in recent years. However, relatively little work has focused on assessing the robustness of these biomarkers, i.e., investigating the uncertainty of the AI/ML models that these biomarkers are based upon. This paper addresses this issue by proposing a framework to evaluate the already-developed classifiers with regard to their robustness by focusing on the variability of the classifiers’ performance and changes in the classifiers’ parameter values using factor analysis and Monte Carlo simulations. Specifically, this work evaluates (1) the importance of a classifier’s input features and (2) the variability of a classifier’s output and model parameter values in response to data perturbations. Additionally, it was found that one can estimate a priori how much replacement noise a classifier can tolerate while still meeting accuracy goals. To illustrate the evaluation framework, six different AI/ML-based biomarkers are developed using commonly used techniques (linear discriminant analysis, support vector machines, random forest, partial-least squares discriminant analysis, logistic regression, and multilayer perceptron) for a metabolomics dataset involving 24 measured metabolites taken from 159 study participants. The framework was able to correctly predict which of the classifiers should be less robust than others without recomputing the classifiers itself, and this prediction was then validated in a detailed analysis.

## 1. Introduction

A biomarker is a quantifiable measurement of a biological sample that is characteristic of a pathological state [1]. However, many conditions cannot be diagnosed by looking at individual quantities. Instead, multiple measurements are needed for diagnosis. This is especially so for most “-omics” data where vast amounts of different quantities can be measured simultaneously [2]. For example, advances in medical research technology such as mass spectrometry have contributed to an increase in the number of metabolites, or products of cell metabolism, that can be measured from a single sample [3,4]. This often results in a complex, high-dimensional dataset with hundreds to thousands of metabolites. As the data become more complex, there is also a need for more advanced analyses to draw conclusions [3].

Often, researchers will use artificial intelligence (AI)/machine learning (ML) algorithms to assess the predictive power of metabolites from the data [3,4] to find a panel of biomarkers. However, as the number of measured metabolites increases, correlations among the metabolites also increase [3]. If the metabolites are highly correlated, it becomes more likely that a classifier based upon these metabolites will overfit the data. Given these reasons, for any identified biomarker based upon machine learning models, there is a growing concern regarding the robustness of biomarker-based diagnostic tests, i.e., that the model will not provide accurate predictions for samples that it was not trained on [4], resulting in inaccurate diagnoses.

This concern is also reflected in a recent announcement by the National Institutes of Health (NIH) dealing with the “validation of digital health and artificial intelligence tools for improved assessment in epidemiological, clinical, and intervention research” [5]. One motivating factor for this notice is that AI/ML biomarker studies often use many different methods, and it is difficult to analyze the efficacy of one over another as, e.g., there is no generally accepted technique to statistically assess how robust these methods are [5,6]. While it is not possible to guarantee the generalizability of any test to new data without collecting such data, the intent is to adopt approaches to predict if a model’s predictions are likely to carry over to data from new clinical trials. 

Traditionally, research pertaining to the selection of optimal AI/ML strategies often foregoes an evaluation of classifier robustness and focuses purely on maximizing the performance (i.e., predictive accuracy) of a classifier. In many of these studies, the differences in classifiers’ performance are often marginal [7,8,9], and little information can be derived regarding the long-term generalizability of such classifiers. To address the problem of robustness, some studies have attempted to use different validation techniques to simultaneously minimize classifier bias and variance [10]. Antonelli et al. suggested that a sensitivity analysis approach to biomarker studies may be required to ensure the robustness of the results [11]. In this regard, the sensitivity of a potential classifier is the degree to which its output accuracy or its parameter values change due to variations in the inputs.

Recent studies have attempted to characterize this variability or sensitivity in classifiers’ performance and parameters using methods known as uncertainty quantification [12,13,14]. While many of these studies have focused on methods that can be applied to estimate the uncertainty of artificial neural network (ANN)-based classifiers, there is comparatively little focus on methods that can be applied to traditional ML classifiers. This is especially important because ANN’s are not necessarily the best algorithm for every application [14]. Furthermore, while the studies that do address the uncertainty quantification of traditional ML techniques focus on so-called instance-level perturbations (subsampling), the impact of feature-level perturbations (adding noise) on the performance of traditional ML classifiers and the variance these classifiers’ outputs is much less explored [7,13]. Therefore, there is an important need to establish a method that can evaluate a classifier’s variability/sensitivity (or lack thereof) in response to feature-level perturbations, especially for traditional ML classifiers, as this can have major implications for the classifier’s robustness.

While a high predictive accuracy is desired for classifiers, the aforementioned findings illustrate that the robustness of the classifiers also needs to be assessed. Measuring only classifiers’ performance ignores that it is possible to obtain a high accuracy for a specific set of data while not performing well on out-of-sample data. In other words, if the perturbations of a dataset of a magnitude that one is likely to encounter in practice will cause significant changes in the classifier parameters, such as the coefficients of a linear model [15], the choice of selected features, or the classifier accuracy, then there is a lack of robustness, and it is unlikely that the AI/ML-based biomarker will perform well on new data. To address this challenge, the contribution of this work is to (a) investigate components of a framework for assessing the robustness of AI/ML-based classifiers and (b) to illustrate these findings on a panel of commonly used classification algorithms that are affected by noise and training/testing split variations of a dataset for one specific example. The goal of this framework is to establish a procedure that can evaluate the robustness of a classifier that has already been constructed from tabular, continuous, and multivariate datasets, such as data derived from metabolomics, proteomics, or transcriptomics. 

The next section (Section 2) introduces the individual components of the framework and how they are used as part of the evaluation procedure. This is then followed by a description of the illustrative example dataset used and details regarding how the classifiers tested in this study were developed. Section 3 outlines the general results of applying the framework to the computed classifiers. Finally, the implications of this study are discussed and reflected upon in Section 4 and Section 5, respectively. This includes a detailed discussion of each individual part of the framework, followed by overall inferences regarding both the usefulness and limitations of this work.

## 2. Materials and Methods

### 2.1. Overview of Methods

The goal of this study was to establish a framework to test if an already-derived classifier from biological data is robust. To build such a framework, two elements were required: (1) a method to test the robustness of features included as inputs to the classifier or identified by a separate feature selection process, and (2) a method that computes the sensitivity/variability of a classifier’s performance and parameters in response to feature-level perturbations. To address (1), a factor analysis procedure is detailed in Section 2.2.1 that determines which measurements of a data set are significant features that have a high potential to be useful for a classifier. In essence, any classifier should only include features that are deemed significant either by itself or in combination with other features. The remainder of Section 2.2 addresses (2) with a Monte Carlo approach that makes use of different types of noise as well as noise levels to compute how the accuracy and model parameter values change in response to noise. A Monte Carlo approach was adopted because it assesses the output of classifiers in response to noise while also limiting the impact of outlier results, i.e., by calculating the average over all trials, and providing a metric to measure classifier sensitivity/uncertainty, which is in turn achieved by calculating the variance of all trials. This study serves as a guided use of these methods. To illustrate the core observations of this framework, the aforementioned methods are applied to an example dataset. The details of this dataset are explained in Section 2.3. Classifiers were constructed by choosing hyperparameters and features using the approaches outlined in Section 2.4. After determining the classifiers to be used, factor analysis was applied to the full dataset to compile a list of significant features. Next, a Monte Carlo approach was utilized such that the feature input data of each classifier was repeatedly perturbed with increasing levels of noise. The average and variance of the classifiers’ performance and parameters were recorded for each perturbation method for discussion in the next sections.

### 2.2. Robustness Framework

#### 2.2.1. Factor Analysis Procedure

Metabolomics studies can utilize datasets with many features; therefore, most studies with high-dimensional data use some form of feature selection. Most robust classifiers will be constructed only using a chosen subset of the original features as inputs. It is imperative, however, to test whether the features included in the classifier, i.e., the features chosen by the feature selection method, are truly statistically meaningful. The purpose of this factor analysis procedure is to use a series of statistical tests to indicate which features are most likely to construct a robust classifier. However, this is just a list of potential features and is not a substitute for feature subset selection, as including every significant feature could result in overfitting if they are too numerous.

It is important to determine if the inputs of a particular classifier are part of the significant features of the data set, as most classifiers only make use of a subset of the available measurements. In this regard, a factor analysis method is employed herein that pares down the original dataset into a small subset of important features. This subset of important features should then be compared to the input features to determine if the input features are statistically meaningful in the data. The workflow for performing the factor analysis is shown in Figure 1 [16]. The approach consists of three main parts: (1) false discovery rate calculation, (2) factor loading clustering, and (3) logistic regression variance calculation. *F′*, *F″*, and *F** are the remaining features after steps (1), (2), and (3), respectively (as per Figure 1). Each step removes features until a subset of features is left, and the factor analysis procedure is focused on achieving certain benchmarks without defining a specific number of metabolites to screen for. 

After standardizing the data, the factor analysis computes the univariate false discovery rate of each variable. For each variable, the *p*-value is computed according to the decision tree, shown in Figure 2, while one sample is left out; then, additional *p*-values for the variable are computed such that two samples at a time are left out. This is repeated until each sample (or pair of samples in the case of leave-two-out) has been left out once. The false discovery rate for each variable is computed as the number of *p*-values calculated to be greater than 0.05 divided by the number of all *p*-values calculated [16]. As such, the false discovery rate of each feature *f* is computed as the following:(1)$$FDR_{f}=\frac{Count\;of\;p\;values>0.05}{Total\;count\;of\;p\;values}$$

All features with FDR greater than 0.1 are removed from further analysis leaving a subset of *m* different features to be considered. Principal Component Analysis (PCA) is performed on the remaining data to attempt to remove redundant or highly correlated variables [17]. This results in a series of principal components of dimension *m*. Selection of the most important principal components can be performed using Horn’s parallel analysis [18].

The contribution of each variable in the data to each principal component can be found by inspecting its corresponding element in the component vector. For example, the first element of each principal component vector marks the contribution of the first feature column in the original data, the second element marks the contribution of the second feature column, and so forth. These can be considered “factor loadings”. As such, the factor loading matrix, *A*, for all original variables can be computed, given that the matrix containing eigenvectors only of the selected principal components, *P*, and the matrix containing eigenvalues of selected principal components on the diagonal, *V*:(2)$$A=PV^{-1/2}$$

K-means clustering is subsequently performed to group the factor loadings [19]. The grouping of the clusters is determined via computing the silhouette score [20]. The silhouette score was chosen as a clustering metric because it is a measure of how tight a cluster is and does not require ground truth labels, as opposed to the adjusted rand index and adjusted mutual information, which both require preexisting labels [20,21,22]. The silhouette score for cluster *c* is computed via Equation (3). Given the average distance between points within the same cluster, *a*, and the average distance from a point within the cluster to a point of the next cluster, *b*:(3)$$s_{c}=\frac{b_{c}-a_{c}}{max\left(a_{c},b_{c}\right)}$$

If the silhouette score for a cluster is sufficiently high, a subset of three variables is taken from that cluster for further analysis. The idea behind this approach is to remove excessively similar variables from a tightly knit cluster [23]. This results in another, smaller set of features.

A logistic regression classifier is trained using the remaining features (*F″*), generating a vector of weights *w* with length equal to the number of remaining features in the feature space. This step ensures that the features chosen are relevant in a classification task. For each remaining feature, the variance of that feature is calculated by replacing all of that feature’s values with random samples from a Gaussian distribution and with values from all other features set to zero. The continuous output is computed by multiplying *w* with the data. The variance of the metabolite is then scaled such that the greatest variance is equal to eight. All features with variance less than two on this scale are considered uninformative and removed from the dataset. This is because a scaled variance of 8 encompasses approximately 99% of all possible values in a normal distribution. In essence, in an N(0,1) distribution, a variance of 8 corresponds to a standard deviation of roughly 2.82, where ~99.76% of data lies within 2.82 standard deviations [24]. All features remaining (*F**) after this analysis can be considered important features for further analysis. When evaluating a classifier for robustness, one of the first steps should be to investigate if the inputs of the classifier are taken from the subsets of important features. 

#### 2.2.2. Noise Corruption

For AI/ML diagnostic tests to be clinically relevant, they should be applicable to data not used for training. To measure a model’s generalizability to other potential data, the existing dataset should be corrupted by some noise factor to determine if similar accuracy can be achieved with modifications applied to the data [25]. Mass spectrometry data are subject to two main sources of noise: random errors and systematic errors [26]. Random errors result from the misinterpretation of faulty signal spikes due to technological errors, and systematic errors refer to the inaccuracies of measurements caused by drift or batch effects. Two different types of noise will be considered here to model these sources of error: (1) replacement noise and (2) Gaussian noise.

Replacement noise refers to the situation where a certain proportion of the data, *p*, is replaced by a random value sampled from the original distribution of the data. As the amount of data replaced by noise increases, the information content in the data set progressively becomes closer to pure random noise. Therefore, it is expected that the performance of any model approaches 50% classification accuracy when the data are completely replaced by noise. The algorithm for applying replacement noise is described in Algorithm 1.
**Algorithm 1:** Replacement Noise AlgorithmInput: Dataset *D* with *N* samples and *F* featuresOutput: *D*’, a corrupted dataset with *N* samples and *F* features1: Define noise term *p*, the proportion of values to replace2: For each f ∈F:3:   $d_{f}=column\;f\;of\;Dataset\;D$4:   $\mu_{f}=\frac{sum\left(d_{f}\right)}{N}$5:   $\sigma_{f}=\sqrt{\frac{(\sum_{n=0}^{N}d_{n,f}-\mu_{f})\;}{N-1}}$6:   v=p∗N7:   Randomly remove *v* values from column $d_{f}$8:   Replace each value with a randomly sampled value from *N*$\left(\mu_{f},\;\sigma_{f}^{2}\right)$9: End for loop

Gaussian noise is additive to existing data and has been used in previous studies to model systematic errors [27]. A random value is drawn from a Gaussian distribution with mean 0 and standard deviation equal to the standard deviation of the variable, σ, multiplied by some dampening factor, *s,* such that the noise term distribution has a standard deviation of σ × *s*. The noise terms are then added to the original data. If low levels of Gaussian noise corruption cause a significant decrease in accuracy or a large variance in a classifier’s performance, it is not likely that the classifier will generalize well to other data. The algorithm for applying Gaussian noise is described in Algorithm 2. In this study, corrupted values that were less than zero were corrected to be zero due to the use of metabolomic concentration data. More details about the dataset can be found in Section 2.3. Dataset.

#### 2.2.3. Severity of Noise

The effect of noise severity on a classifier can be investigated for both replacement and Gaussian noise by adjusting tunable factors. For replacement noise, the proportion of values replaced can be adjusted. In this study, the proportion of data points replaced, *p,* was adjusted from 0% to 100% of the dataset in increments of 10%. In the instance of Gaussian noise, the dampening factor, *s,* was adjusted from 0 to 1 in increments of 0.1. At each value of the noise level (value of *p* or *s*), the dataset was corrupted 1000 different times. It is important to note that the data are corrupted before standardization. The data were then split into a training and a testing set. The same samples (patients) were contained in the training and testing set for each of the 1000 corruptions to maintain consistency in the sampled noise distribution. Given a specific classification algorithm, the parameters of the classifier were trained using the training set. The mean Leave-One-Out Cross-Validation (LOOCV) accuracy of the training set, the mean testing accuracy, and the values of the model parameters, as well as the variance of these values, were computed over the 1000 corruptions. This process was repeated for all noise levels under investigation.
**Algorithm 2:** Gaussian Noise AlgorithmInput: Dataset *D* with *N* samples and *F* featuresOutput: Noisy dataset *D*’ with *N* samples and *F* features1: Define noise term *s*, which scales the standard deviation of the noise distribution $x_{f}$2: For each f ∈F:3:   $d_{f}=column\;f\;of\;Dataset\;D$4:   $\mu_{f}=\frac{sum\left(d_{f}\right)}{N}$5:   $\sigma_{f}=\sqrt{\frac{\{\sum_{n=0}^{N}d_{n,f}-\mu_{f})\;}{N-1}}$6:   $q=s*\sigma_{f}$7:   $x_{f}\sim N\left(0,q^{2}\right)$8:   $d_{f}^{'}=d_{f}+x_{f}$9: End for loop


#### 2.2.4. Monte Carlo Data Splitting

While metabolite concentration variations can be assessed by perturbing the dataset with noise, it is also important to investigate different training/testing set splits to ensure that results are not just representative of one split [28]. First, the dataset is split many times while maintaining the ratio of 70% of the data in the training set and 30% of the data in the testing set. For this work, the data were randomly split 1000 times with no additional noise. Then, the testing accuracy and LOOCV training accuracy were computed for each split. The average and variance of these performance metrics were computed across the 1000 splits to determine if the training set and testing set result in similar accuracies for any given training/testing split. Additionally, the model parameters were recorded for each split that is followed by computation of the average and variance of the parameters to determine if the model structure changes for different training/testing splits.

### 2.3. Dataset

Since a robustness framework will also have to be tested on specific classifiers, this study makes use of one specific data set. This data set contains metabolite concentrations from children with an autism spectrum disorder (ASD) diagnosis and their typically developing peers. Autism is a disorder that currently has no generally accepted biomarker as the current method of diagnosis is purely by observation [29]. Given the complexity involved in screening young children by observation only, availability of a physiological test to support a diagnosis would be desirable. The dataset used in this analysis is derived from 159 blood samples taken from children aged 3 to 10 [30]. Of these 159 samples, 83 were from children with an autism diagnosis and 76 were from typically developing children. Furthermore, both groups of children were age matched. For each blood sample, 24 metabolites from the folate one-carbon metabolism and the transsulfuration pathways were measured. These metabolites were analyzed using High-Performance Liquid Chromatography (HPLC). Internal standards were used to control the batch effects present between runs. A comprehensive list of the analyzed metabolites is provided in Howsmon et al.’s paper [30]. The ASD diagnosis was verified via the Autism Diagnostic Observation Schedule of Childhood Autism Rating Scales [30]. Typically developing children had no medical history of neurologic symptoms reported by the parents [30]. The data originated from the Arkansas Children’s Hospital Research Institute autism IMAGE study, which was approved by the Institutional Review Board at the University of Arkansas for Medical Sciences [30].

### 2.4. Classifiers

#### 2.4.1. Classifier Overview

To assess the validity of the presented framework, five commonly used traditional ML algorithms and one neural network were used to develop classifiers for the dataset from Section 2.3 and evaluated using the steps described in Section 2.2. The goal of evaluating multiple classifiers is not to compare the outright performance between algorithms, but rather to demonstrate that different classifiers for the same dataset may not all be equally robust. Furthermore, it is useful to illustrate that this framework can be applied to classifiers constructed using different AI/ML algorithms. Specifically, the ML techniques used were (1) linear discriminant analysis (LDA) [31], (2) support vector machine (SVM) [32], (3) random forest (RF) [33,34], (4) partial least squares-discriminant analysis (PLS-DA) [35,36], and (5) logistic regression (LOG) [37,38]. While the primary focus of this study is to develop a robustness evaluation framework for traditional classifiers, a multilayer perceptron (MLP) was also used as many multivariable biomarkers make use of neural networks [39]. More details regarding each of these algorithms can be found in Section 2.4.2. Each of these methods was implemented using the appropriate functions of the scikit-learn package in Python [40]. Only five features were used as inputs for each classifier to prevent overfitting. A brief description of the feature selection technique is included in this section. 

#### 2.4.2. Classification Algorithms

##### Linear Discriminant Analysis

Linear Discriminant Analysis (LDA), also called Fisher Discriminant Analysis, aims to determine the greatest separation between two classes by fitting a vector (or component) that maximizes the between-class variance, while minimizing the within-class variance [31]. LDA is a latent variable method, as the vector is not a variable that is directly observed but is synthesized by contributions of the observed variables. The values from the multivariate dataset are then projected onto this vector, which classifies by defining a threshold as the decision boundary [31].

##### Support Vector Machine

Support Vector Machine (SVM) is a quadratic programming algorithm that aims to fit a classification boundary such that the width of two vectors parallel to the boundary, denoted as margins, have the maximum distance between them [32]. The location of these margins is characterized by the closest point to the potential decision boundary. In the likely event that the data are not perfectly linearly separable, a regularization term that allows some of the data to penetrate the decision boundary can be used, with a higher regularization term allowing for less freedom for the data to penetrate the boundary [32].

##### Random Forest

Random Forest (RF) is a collection of randomly initialized decision trees that are each fitted according to the CART algorithm. Effectively, each tree in the forest is initialized with a node, thresholding a variable and subsequently splitting the data into two groups [33]. This spawns two daughter nodes, where each node represents a threshold in the data. While RF borrows its tree growth from the CART algorithm, it differs in two important ways. Each tree in the RF is fit using a randomly selected bag of samples, each selected with replacement from the original sample space [33]. Rather than exhaustively searching the feature space for thresholds, the feature used to threshold at the next node is randomly selected. This results in many uncorrelated models to compare and the most unbiased estimate from the forest. The threshold is based off the threshold that minimizes the Gini impurity [33,34]. The Gini impurity is a measure of the probability that a datapoint is classified incorrectly by the tree. Due to the nonlinear nature of an RF model, there are no coefficients that can be used to estimate the relative importance of a feature in the prediction. However, the average decrease in the Gini impurity across all trees for each feature is representative of that feature’s importance in the model [34]. To obtain a resultant prediction, each tree predicts the sample’s class membership, and the majority vote is taken as the overall classifier prediction.

##### Partial Least Squares-Discriminant Analysis

Partial Least Squares-Discriminant Analysis (PLS-DA) is an extension of the Partial Least Squares (PLS) regression algorithm. Similar to LDA, PLS-DA is a latent variable method, making it suitable for high-dimensionality datasets because it can distill information from many variables into one vector [35]. Given a data matrix and a response variable vector, the PLS-DA algorithm finds the vector that maximizes the covariance between the data matrix and the response. PLS-DA inherently computes a continuous output of values on the fitted component, so a threshold is defined (for this study, 0.5) that sets all values above it to one classification outcome, and all values below the threshold as the other classification outcome [36].

##### Logistic Regression

Logistic Regression is an iterative method that, given a data matrix and response variable, attempts to find a vector of weights that—when multiplied with a vector containing the data of one sample—identifies the correct class of that sample [37]. A random vector of weights is first initialized and is then iteratively updated to minimize the ‘loss’ function, i.e., the error of the output. This optimization problem is solved using the SAGA solver, which is an adaptation of stochastic average gradient descent [38]. The SAGA variant of gradient descent was chosen because it performed the best on the example dataset presented in the study. Newton’s method or Limited-Memory BFGS are usually the best choices of optimization algorithm for small datasets [40], but in this specific case SAGA reached higher accuracy with insignificant differences in convergence time. Additionally, SAG-based methods also provide faster runtime as the dataset becomes larger, with the SAGA variant usually being faster than standard SAG. 

##### Multilayer Perceptron

A multilayer perceptron (MLP) is one type of artificial neural network. Input data, i.e., features, are connected to outputs, i.e., classification outcomes, through a series of nonlinear transformations stored as hidden layers [39]. Every hidden layer is composed of a user-specified number of neurons, which are transformations of the data from the layer before it. These data are transformed by multiplying the data in the previous layer by a set of weights that are computed iteratively to minimize a loss function. The architecture used in this study is a fully-connected neural network with two hidden layers and 32 neurons in the first hidden layer and 16 neurons in the second hidden layer. The output layer uses an activation function to threshold the data in order to provide binary information for the classification.

#### 2.4.3. Classifier Construction

Classifiers’ hyperparameters were established by splitting the data into a 70% training and 30% validation set split and using a grid search technique among a set of potential parameters. Optimal hyperparameters were determined by the set of parameters that yielded the greatest LOOCV accuracy. These hyperparameters are listed in Appendix A.

Feature selection was performed using a wrapper method. Wrapper feature selection requires evaluating a trained classifier’s performance metric, where the classifier is trained using a subset of features [41]. For every combination of five metabolites (five was specifically chosen based on Howsmon et al.’s and Qureshi et al.’s original analyses), the data are split into the same 70% training and 30% testing set. The optimal feature subset (for each algorithm) is selected based on the best LOOCV accuracy of the training set. A total of 5 metabolites were chosen because 6 or more features did not significantly improve accuracy and increased the risk of overfitting by providing too many features. Capping the number of input features at 5 maximizes the classifiers’ performance for this dataset while also minimizing potential overfitting concerns. This is also in line with the findings from Howsmon’s [30] and Qureshi’s study [42]. To validate the results of this feature selection, a similar procedure was carried out where the data were split the same way 1000 times and the combinations yielding the highest testing accuracy from each split were saved. Those features were then ranked by “prevalence”—the proportion of combinations that feature was in. Feature subsets with lower accuracy but with greater prevalence among (top 1/3rd) the features were considered over feature subsets with high accuracy but less prevalent features.

The feature subsets selected this way for each classifier were evaluated for their robustness by comparing them to the features identified as statistically significant by the factor analysis described in Section 2.2.1.

## 3. Results

The first step of the robustness assessment was to compare the features identified as statistically significant by the factor analysis to those that were selected as inputs for each classifier. A deeper analysis of the feature selection results is present in Appendix C. A total of 11 metabolites of the original 24 were identified as significant by the factor analysis protocol. These metabolites were % DNA methylation, 8-OHG, fCystine/fCysteine, Nitrotyrosine, Tyrosine, Glu-Cys, Cys-Gly, GSSG, tGSH/GSSG, %Oxidized Glutathione, and Chlorotyrosine. All 7 metabolites identified by Howsmon et al.’s classification analysis were present among these 11 metabolites. Using that panel of seven metabolites, Howsmon was able to achieve a cross-validatory accuracy of 96.9% using LDA [30]. In a more recent study using the same dataset, Qureshi et al. was able to achieve a cross-validatory accuracy of 97% using SVM and five metabolites [42]. Again, all features used in this classifier were present among the 11 identified by the factor analysis. The features selected as inputs for each classifier in this study are included in Table 1. For LDA, MLP, and LOG, all the features used as inputs were among the metabolites deemed important by the factor analysis protocol. In the case of PLS-DA and SVM, four out of five were identified as important (% DNA Methylation, 8-OHG, Glu-Cys, and fCystine/fCysteine), whereas in the RF classifier only two were considered important (%DNA Methylation and fCystine/fCysteine). Therefore, it is expected that the RF classifier is more sensitive to variations than the other classifiers due to the fact that it contains less statistically relevant features. To a lesser degree, these concerns also exist for the SVM and PLS classifiers.

After assessing the quality of the features used as inputs for the classifiers, the behavior of the classifiers in response to noise was analyzed. A summary of the average accuracy and sensitivity, i.e., the variance of the accuracy, in response to the replacement noise, Gaussian noise, and varying training/testing split is shown in Table 2. For the uncorrupted data, each algorithm results in a similarly high accuracy. In response to replacement noise, the testing accuracy followed an approximately linear trend from the original classifier with an above 90% accuracy at *p* = 0, i.e., no values replaced, to approximately random chance at *p* = 1, i.e., all the values replaced, resulting in an approximately 50% classification accuracy. This represents an approximately 4% accuracy decrease per 10% of values replaced. This trend is displayed clearly in Figure 3 for all the classifiers. The LOOCV training accuracy was found to exhibit the same trends as the testing accuracy, and the corresponding Figure A1 can be found in Appendix B. Figure 4 shows the variance of the testing accuracy, which increases approximately linearly with the replacement noise (corresponding Figure A2). Overall, this behavior is expected as the dataset is being progressively replaced by random noise until it is just random noise, at which point no classifier should be able to discriminate well.

The change in the feature importance in response to replacement noise was assessed by the change in the average classifier parameter values with an increasing noise level *p* (Figure 5) and the variance of these parameters across 1000 noise corruptions of the same value of *p* (Figure 6). All the parameters effectively start at separate and distinct values, and they all converge to become approximately the same as *p* increases. Essentially, as more data are replaced by random noise, the differences between the information in each feature is less pronounced. For each classifier, the parameters tend to follow the same trends for the parameter variance with an increasing *p*. It should be noted that the parameters that had a greater overall increase in variance and demonstrated such an increase at a lower proportion of replaced values were the parameters of RF that were not identified by the factor analysis procedure, including the coefficients of Methionine, tGSH, and fGSH/GSSG. However, in the PLS and SVM classifiers, the parameters of the feature not identified by the factor analysis procedure (fGSH/GSSG) did not deviate significantly from the behavior of the other parameters.

While Gaussian noise corruption caused a decrease in classifier accuracy in response to increasing noise level(s), this decrease was less severe than the accuracy decreases due to the replacement noise. Similar trends for the classifier accuracy are present for the testing accuracy and LOOCV accuracy, which are shown in Figure 7 and Figure A3, respectively. Figure 8 shows that the variance in the classifiers’ performance increases marginally for the LDA, MLP, SVM, and LOG classifiers in response to Gaussian noise (Figure A4 for cross validation). However, the variance of the performance increases significantly with increasing Gaussian noise levels for the RF classifier, and to a lesser extent the PLS-DA classifier. This further validates the idea that these classifiers are more sensitive to noise. Conversely to replacement noise, the classifier parameters were largely unchanged in response to Gaussian noise, shown in Figure 9. The variance of the parameters (Figure 10) still increased with respect to the Gaussian noise level, s; however, a trend as to which parameters varied the most for each classifier was not apparent.

The last step of the evaluation procedure was to examine the effect of varying the training/testing split on the performance of the classifier, which is visualized in Figure 11 and summarized in Table 2. There was no significant difference between the average testing accuracy and the LOOCV training accuracy across the 1000 training/testing splits for any classifier. The testing and LOOCV accuracies were computed to be, on average, the lowest for the RF classifier (92.0% accuracy and 92.4% accuracy, respectively) among the six classifiers investigated. This is significantly less than the average accuracies for all the other classifiers across the 1000 testing splits (*p* < 0.05). For all the cases, the variance of the LOOCV training accuracy was significantly lower than the variance of the testing accuracy for each classifier. This indicates that using LOOCV is less of a stress test than choosing different testing/training sets, yet there are obvious advantages to using LOOCV for small data sets. The performance variance was relatively low for all classifiers except for RF, further raising doubts about the robustness of the determined RF-based classifier. The accuracy of this classifier was maximally computed to be 100% and minimally computed to be 77.1%, corresponding to a 12.5% accuracy variance. Comparatively, no other classifier-computed accuracies were below 85% for any training/testing split or had a variance greater than 8.5%. The connection between the variance in the performance and robustness is investigated further in the Discussion section. 

## 4. Discussion

The major motivation for this work was to present a robustness evaluation framework for AI/ML-based classifiers. This proposed framework was then subsequently used to assess the robustness of several classifiers derived using different methods. A detailed discussion resulting from this analysis is presented in this Section. 

To determine if a classifier is robust, one must first determine if the features included as inputs to a given classifier will be sensitive to changes in the data. This was carried out via factor analysis, which identified 11 metabolites as statistically meaningful for the investigated data set. For a classifier to be robust, most or all of the features used to develop the classifier should be identified by the factor analysis approach as being impactful in the data. This finding becomes apparent as three of the six investigated classifiers (LDA, MLP, LOG) have all five of their inputs selected from the set of important features and show reasonably low variations in accuracy in response to the replacement noise and Gaussian noise. Random Forest, alternatively, had only two of five inputs come from the list of important features and its variance was significantly larger than that of the other algorithms. For example, RF exhibited the greatest variance in accuracy in response to replacement noise (Figure 4), Gaussian noise (Figure 8), and the random alteration of the training/testing split (Figure 11). As such, when applying these approaches to future classifiers, one should be aware that a high variance in the classifier outputs may indicate that a classifier’s features are overly sensitive to perturbations.

Another finding of this work is that it is possible to assess how much random noise a classifier can handle before it is no longer considered robust. The classifiers’ performance diminished approximately linearly after corrupting a dataset with progressively harsher replacement noise, until the accuracy equates to random chance, as illustrated in Figure 3. For example, the testing accuracy for this dataset starts out at over 90% for all algorithms and continuously decreases by approximately 4% accuracy per 10% of the data replaced. At a 100% data replacement rate, all the algorithms end up at approximately 50% accuracy. This behavior was expected for each classifier because if the dataset that a classifier is based upon consists solely of random noise then the classifier should predict correctly approximately 50% of the time. In this regard, none of the classifiers had significantly different trends for the accuracy in response to increasing levels of replacement noise. For clinical significance, an accuracy of at least 80% is often assumed. Given that one can compute the accuracy for the nominal dataset, and it is shown that the accuracy decreases linearly to 50% at 100% replacement noise, one can estimate how much random error can be in a dataset to still obtain meaningful results. 

This study also examined the robustness of classifiers to systematic errors. Systematic errors in metabolomics data often take the form of small perturbations of the existing data and have been previously modeled using Gaussian noise [26,27]. Therefore, it would be desirable for a classifier to maintain a high performance and relatively unchanging parameter values due to Gaussian noise. This was found for most of the classifiers, as the mean and variance of the accuracy did not change much for the LDA, MLP, SVM, and LOG classifiers, as can be seen in Figure 5 and Figure 6, indicating that their performance is not as sensitive to Gaussian noise as the PLS-DA and RF classifiers, which showed noticeable decreases in accuracy and greater variance. For RF, this can be explained by the fact that individual decision tree classifiers are known to overfit to noise. While most RF classifiers use a sufficient number of trees to mitigate overfitting, the RF classifier used in this study uses very few (16) to decrease computational load. On the other hand, PLS-DA is based on a threshold, and a high severity of Gaussian noise can cause frequent penetration of this boundary (especially at *s* > 0.4). Notably, both of these classifiers, i.e., PLS-DA and RF, are classifiers for which the robustness was at least partially a concern due to the discrepancy between the classifier input features and statistically meaningful features determined by the factor analysis. 

When assessing the impact of using many training/testing splits, varying the training/testing split is an important aspect for determining the generalizability of a classifier to new data. Essentially, a classifier will likely perform poorly on new data if the classifier’s performance greatly changes by just changing which samples are in the training set. While the LOOCV accuracy was not significantly different from the testing accuracy for any classifier, the variance of the LOOCV training performance for each classifier was significantly lower than the testing performance variance, shown in Figure 11 and summarized in Table 2. This indicates that using different training/testing sets to evaluate the robustness of performance is a more stringent requirement than using LOOCV. This observation is along the lines of what was mentioned by Vabalas et al., who commented that training/testing set performance is less biased than K-Fold Cross Validation performance [10]. That being said, for small sample sizes, LOOCV may be more precise than generating separate training/testing sets as these sets would contain too few data points. 

By observing the behaviors of classifier parameters in response to noise, one can gain an understanding of how sensitive the classifier is to that source of noise. While maintaining the same average parameter values in response to noise is important, the variance of the values of the classifier parameters should also be used to consider a classifier’s robustness. If the parameter variance is high, this indicates that the classifier is overfitting noise as the classifier parameters are changing frequently due to noise corruption. The parameters in the RF classifier corresponding to the features not identified as significant by the factor analysis showed higher variance in response to the replacement noise than the other parameters (Figure 6C), thus exhibiting an increased sensitivity to replacement noise. While the classifier parameters of RF changed significantly in response to replacement noise (Figure 5), the values of the same parameters show much less variation due to Gaussian noise (Figure 9). Similarly, the variances of the parameter values in response to replacement noise (Figure 6) was greater than those in response to Gaussian noise (Figure 10), indicating that the classifier parameters values change less frequently in response to Gaussian noise than replacement noise. This is consistent with the expectation that replacement noise is a more severe form of perturbation of a dataset than Gaussian noise. 

While it has been demonstrated that the developed RF classifier (and in some respects the PLS classifier) is less robust than the other classifiers, it should be emphasized that the purpose of this study was not to find the optimal classifier from a suite of classifiers. Rather, the goal was to demonstrate the use of the proposed robustness evaluation framework by comparing several different classifiers. Using this framework, one can state whether a developed classifier is reasonably robust, but not necessarily if one method for developing a classifier is superior to others. For instance, in Table 2, all the algorithms perform similarly towards the dataset with regard to their prediction accuracy. However, even classifiers that achieve the same performance under a no-noise scenario, such as LDA and PLS-DA, have been demonstrated to perform differently when the dataset is corrupted by noise. Therefore, using a Monte Carlo perturbation-based approach to evaluate robustness can serve as a validation step for classifiers generated from tabular biomarker data. Using these methods, this discussion has identified the RF classifier in this study as being less robust than the other classifiers. This does not necessarily make RF a worse classification method than the others, but the RF classifier developed for this particular dataset was demonstrably less robust than the classifiers developed using other methods. On the other hand, the LDA, SVM, and logistic regression classifiers were shown to be robust to the perturbation methods in the framework. The developed method is generally intended for use on biological datasets such as RNA expressions, metabolite concentrations, or protein concentration data because these types of datasets involve similar noise characteristics to the ones investigated here [43,44,45]. 

Nevertheless, this work also has some limitations that could be addressed in future research. First, using an MLP classifier in this study demonstrated that this framework can only draw limited conclusions from classifiers without clear feature weights or feature importance measures. Second, this framework is not directly applicable to non-tabular data such as imaging data [46]. Therefore, the feature weight of a classifier constructed from imaging data will not yield the same interpretability as for metabolomics or proteomics data. Additionally, the types of noise that are applied to imaging data extend far beyond Gaussian noise and can also depend on the imaging modality (MRI, CT, etc.) [47].

## 5. Conclusions

The aim of this study was to propose components of a framework that can be used for evaluating the robustness of classifiers. In short, if one is given a dataset and an already developed classifier, this work investigated how one goes about determining whether the classifier is indeed robust, with particular attention to AI/ML-based classifiers. Since this is not a question that can be universally answered, a significant effort was put into the development and evaluation of six commonly used AI/ML algorithm-based classifiers given a variety of different perturbations of a metabolomics dataset for supporting autism diagnoses. These classifiers were not directly compared; rather, they were independently assessed with regard to their robustness.

This framework first performed a factor analysis to determine which features are significant and then evaluated if the inputs used in the classifier form a subset of these important features. Then, it was shown in a case study that classifiers that met this criterion had lower variance in their performance than, e.g., Random Forest, which had only two of its five inputs come from this set of significant features. Another finding is that if a dataset contains replacement noise, then the model prediction accuracy degrades linearly from its best value (at no replacement noise) to 50% accuracy when the entire dataset consists of replacement noise. This information can be used to determine which level of replacement noise a classifier can tolerate to remain above a certain accuracy threshold, i.e., it is often 80% for diagnostic purposes. Monte Carlo simulations involving Gaussian noise and variations of the training/testing split also form important aspects that need to be investigated for assessing classifiers’ robustness. In general, the results from this investigation indicate that replacement noise is more demanding for classifier robustness as compared to Gaussian noise and the same holds true for varying training/test splits as compared to leave-one-out cross-validation.

Further research should take into account which types of noise are applicable to different domains and determine if this framework should be adapted for those domains. For example, replacement and Gaussian noise are realistic for metabolomics data, where there are random and systematic errors, but may not be as applicable to imaging or non-biological domains. In addition, neural networks are commonly used, but it is challenging to assess the impact of uncertainties in specific weights or features due to the nature of connected hidden layers. As such, research on how to better understand these black-box models should be performed in the future.

## Figures and Tables

**Figure 1 jpm-12-01314-f001:**
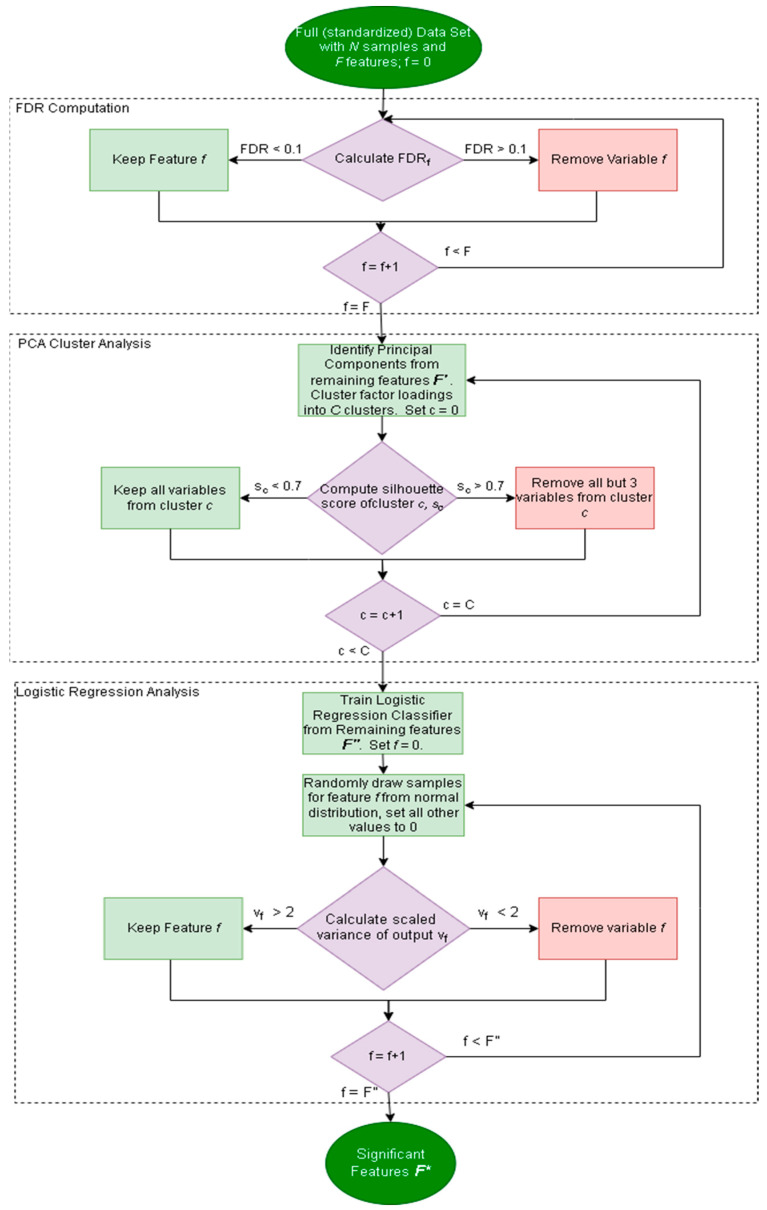
Flowchart describing the factor analysis protocol. Dark green ovals represent the beginning and end of the protocol. Purple diamonds are decision points, light green rectangles are steps where no features are removed, and red rectangles refer to steps where features are removed.

**Figure 2 jpm-12-01314-f002:**
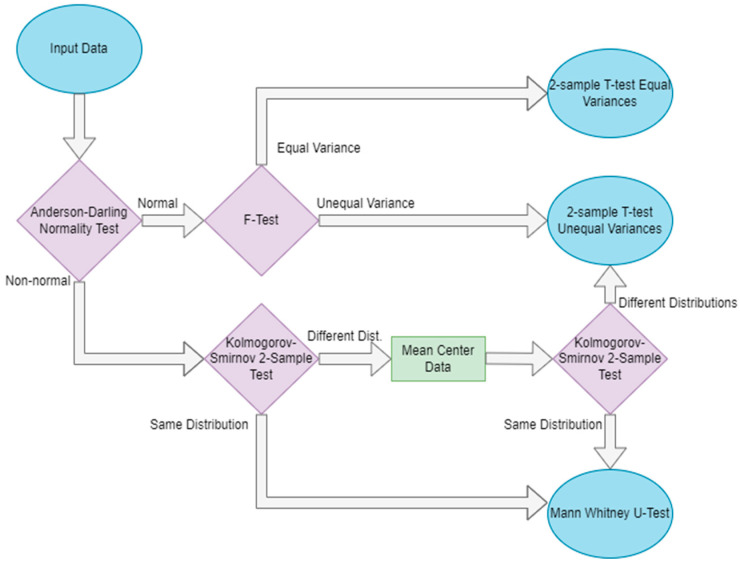
Decision protocol determining which significance test should be used to determine the *p*-value of a variable when computing the FDR. Blue ovals indicate the start and end of the flowchart, purple diamonds indicate decision points, and green rectangles represent intermediate steps.

**Figure 3 jpm-12-01314-f003:**
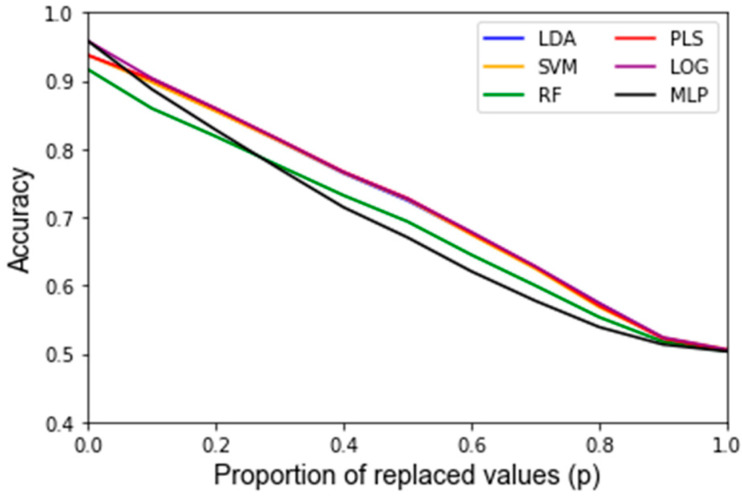
Testing accuracy of replacement noise varying from 0% (*p* = 0) of the data replaced to 100% (*p* = 1) in 10% intervals. Data are the average of classifiers’ performance over 1000 Monte Carlo noise corruptions on the same data.

**Figure 4 jpm-12-01314-f004:**
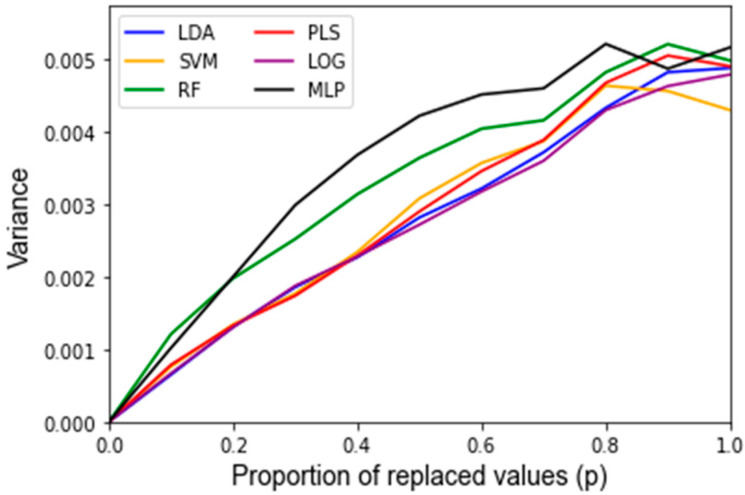
Sensitivity of the testing accuracy of classifiers in response to replacement noise varying from 0% (*p* = 0) of the data replaced to 100% (*p* = 1) in 10% intervals. Data are the variance of classifiers’ performance over 1000 Monte Carlo noise corruptions on the same data.

**Figure 5 jpm-12-01314-f005:**
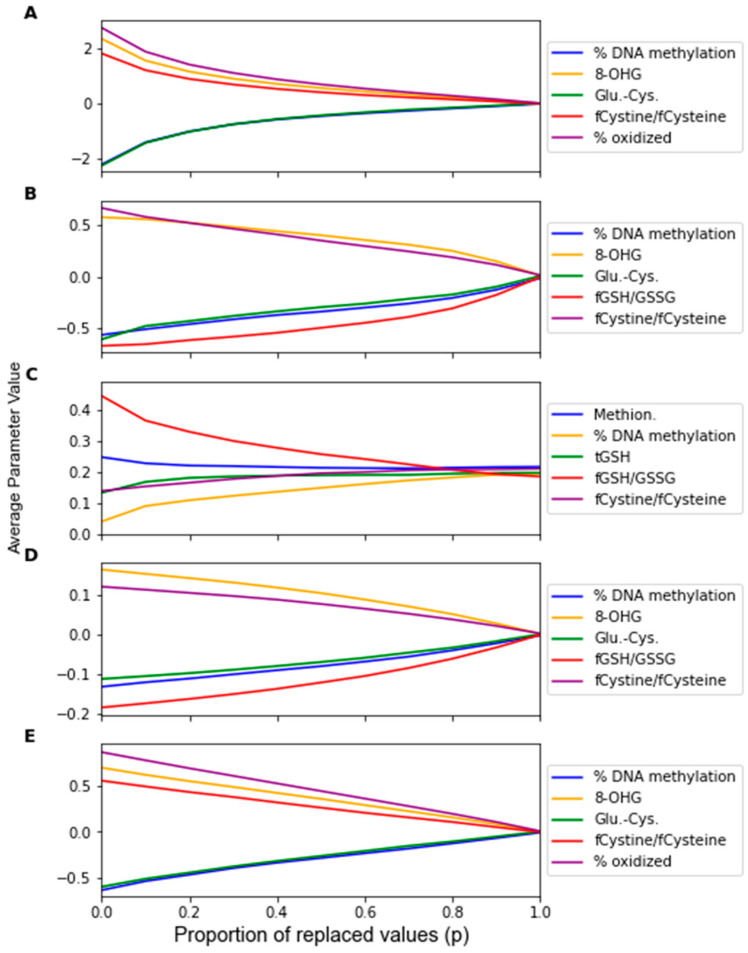
Average classifier parameter values over 1000 replacement noise corruptions at increasing noise level *p*. Noise level was varied from 0 to 1.0 in steps of 0.1. Parameter values were measured for (**A**) LDA, (**B**) SVM, (**C**) RF, (**D**) PLS, and (**E**) LOG.

**Figure 6 jpm-12-01314-f006:**
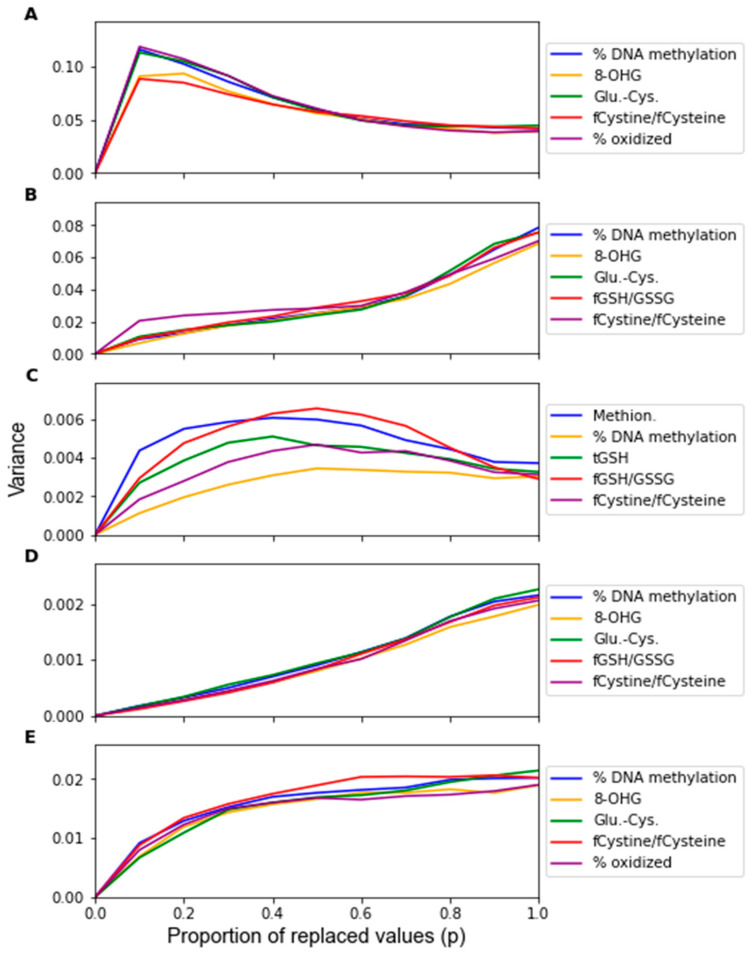
Sensitivity of the parameters as given by the variance of the parameters over 1000 replacement noise corruptions of increasing noise level *p*. Noise level was varied from 0 to 1.0 in intervals of 0.1. Parameter sensitivity was measured for classifiers generated by (**A**) LDA, (**B**) SVM, (**C**) RF, (**D**) PLS, and (**E**) LOG.

**Figure 7 jpm-12-01314-f007:**
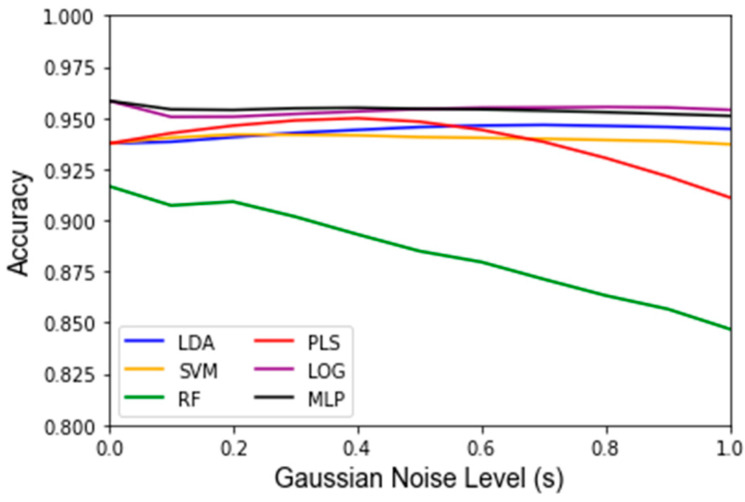
Testing accuracy under influence of Gaussian noise and varying of the standard deviation of the noise term *s* from 0 to 1 in intervals of 0.1. Data are the average classifier accuracies over 1000 Monte Carlo noise corruptions on the same data.

**Figure 8 jpm-12-01314-f008:**
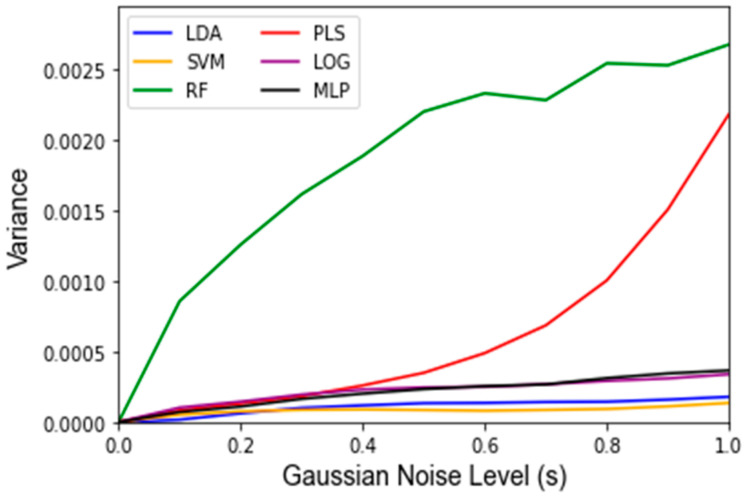
Variance of testing accuracy under influence of Gaussian noise and varying of the standard deviation of the noise term *s* from 0 to 1 in intervals of 0.1. Data are represented as the variance classifier accuracy over 1000 Monte Carlo noise corruptions on the same data.

**Figure 9 jpm-12-01314-f009:**
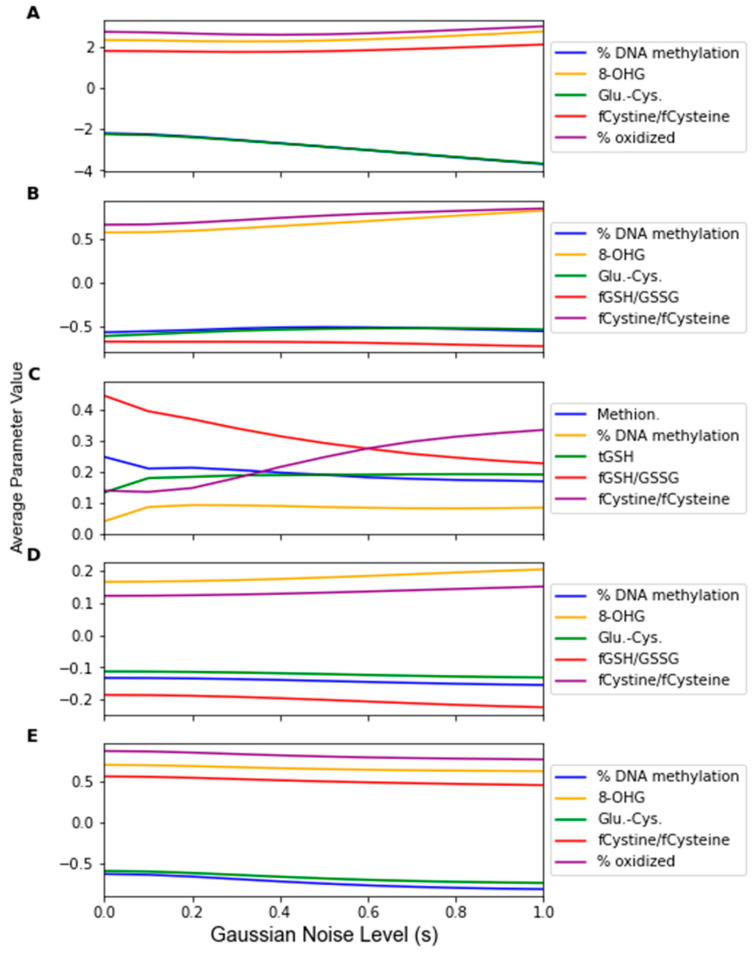
Average classifier parameter values over 1000 Gaussian Monte Carlo noise corruptions at increasing noise level *s*. Noise level was varied from 0 to 1.0 in steps of 0.1. Parameter values were measured for (**A**) LDA, (**B**) SVM, (**C**) RF, (**D**) PLS, and (E) LOG.

**Figure 10 jpm-12-01314-f010:**
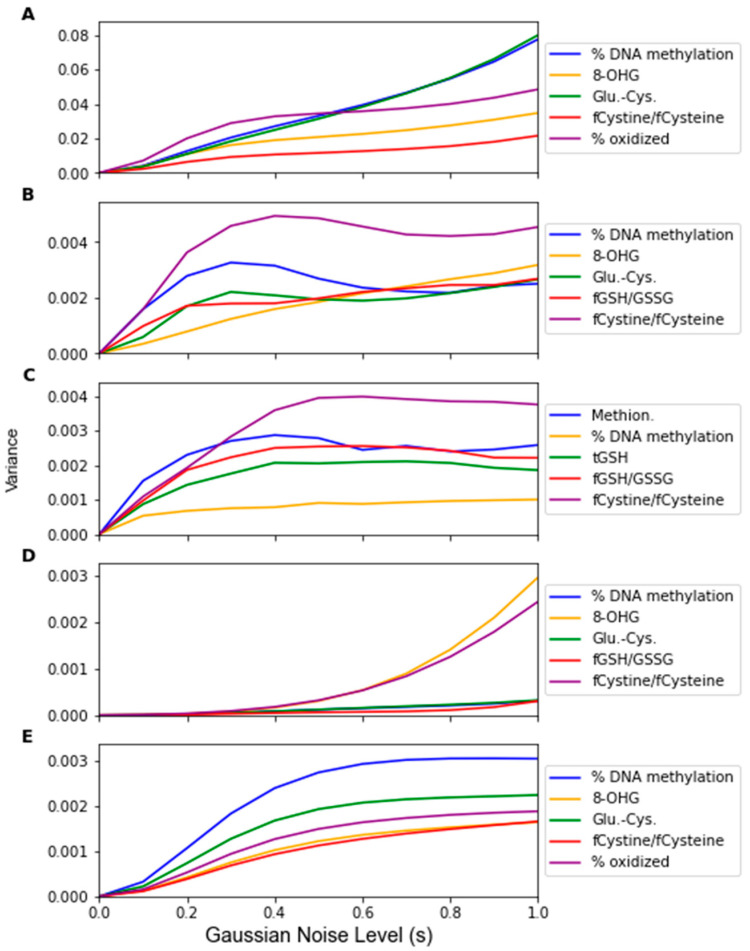
Variance of classifiers’ parameter values over 1000 Gaussian Monte Carlo noise corruptions at increasing noise level *s*. Noise level was varied from 0 to 1.0 in steps of 0.1. Parameter values were measured for (**A**) LDA, (**B**) SVM, (**C**) RF, (**D**) PLS, and (E) LOG.

**Figure 11 jpm-12-01314-f011:**
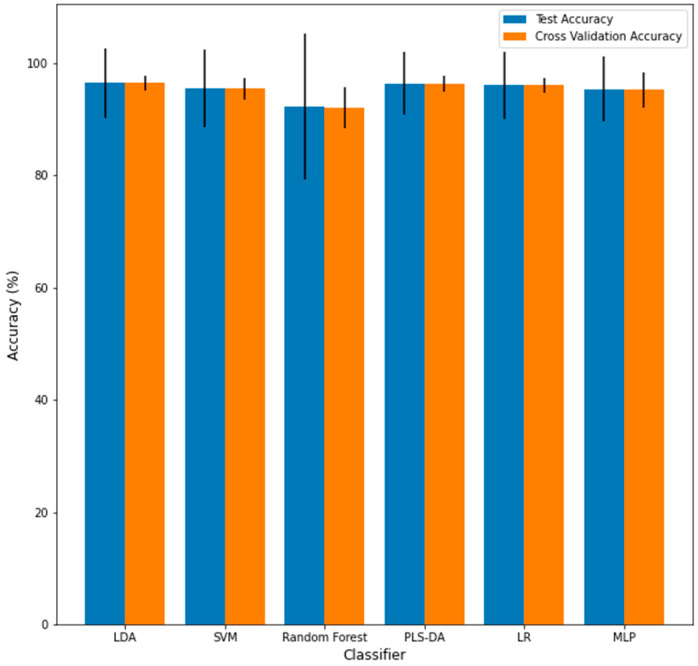
Mean of the testing accuracy (blue) and cross-validation accuracy (orange) across 1000 train/test splits for each algorithm. Error bars represent the variances.

**Table 1 jpm-12-01314-t001:** Features included as inputs for each classifier. Green text features were included in Factor Analysis (F.A.).

Algorithm	Feature 1	Feature 2	Feature 3	Feature 4	Feature 5	# From F.A.
LDA	% DNA Meth.	8-OHG	Glu-Cys	fCystine/fCysteine	% oxid. Glut.	5
SVM	% DNA Meth.	8-OHG	Glu-Cys	fGSH/GSSG	fCystine/fCysteine	4
RF	Methionine	% DNA Meth.	tGSH	fGSH/GSSG	fCystine/fCysteine	2
PLS-DA	% DNA Meth.	8-OHG	Glu-Cys	fGSH/GSSG	fCystine/fCysteine	4
LOG	% DNA Meth.	8-OHG	Glu-Cys	fCystine/fCysteine	% oxid. Glut.	5
MLP	% DNA Meth.	8-OHG	Glu-Cys	fCystine/fCysteine	% oxid. Glut.	5

**Table 2 jpm-12-01314-t002:** Summary of testing accuracy average and variance results.

Classifier	Accuracy No Noise	Accuracy *p* = 1	Variance *p = 1*	Accuracy *s* = 1	Variance *s* = 1	Accuracy MC Split	Variance MC Split	Runtime MC Split
LDA	93.8%	50.4%	4.88 × 10^−3^	94.4%	1.78 × 10^−4^	96.4%	1.40 × 10^−4^	2.80 min
SVM	93.8%	50.6%	4.30 × 10^−3^	93.8%	1.35 × 10^−4^	95.4%	1.93 × 10^−4^	2.77 min
RF	91.7%	50.4%	4.99 × 10^−3^	84.7%	2.67 × 10^−3^	92.0%	3.71 × 10^−4^	36.95 min
PLS-DA	93.8%	50.5%	4.91 × 10^−3^	91.1%	2.18 × 10^−3^	96.3%	1.43 × 10^−4^	2.56 min
LOG	95.8%	50.6%	4.79 × 10^−3^	95.4%	3.38 × 10^−4^	96.0%	1.33 × 10^−4^	3.46 min
MLP	95.8%	50.3%	5.17 × 10^−3^	95.1%	3.65 × 10^−4^	95.2%	3.13 × 10^−4^	46.60 min

## Data Availability

The data presented in this study are publicly available at https://journals.plos.org/ploscompbiol/article?id=10.1371/journal.pcbi.1005385#pcbi.1005385.ref038. accessed on 1 April 2021.

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
