# Peer review of "Framework for Testing Robustness of Machine Learning-Based Classifiers"

_jpm, 2022, doi:10.3390/jpm12081314_

Round 1

Reviewer 1 Report

This paper presents a general framework for assessing the robustness of different machine learning based classifiers in biomedical applications. The framework assesses the robustness of a classifier by examining three different aspects systematically: replacement noise, Gaussion noise addition, and different split of training and testing data. The framework was demonstrated using a publicly available metabolomic data set on Autism, and examined 6 different ML methods (LDA, SVM, RF, PLS-DA, LOG and MLP). The paper is well-written with clearly explained logics. only minor clarification is needed.

1.       The number of the features included in each classifier is capped at 5 based on a prior work. However, different classifiers may need different number of input features to achieve its optimal performance, which could potentially affect its robustness. It would be helpful if the authors can comment on the potential impact of capping the number of input features for all classifiers at 5.

2.       It is interesting to see the different impacts of noise replacement and adding Gaussian noise. For noise replacement, PLS performs similarly to LDA; however, with Gaussian noise added, PLS performs a lot worse when the noise level is high (s > 0.4). Similarly, MLP performed worse for noise replacement but much better for added Gaussian noise. It would be helpful if the authors can comment on the potential reasons for the observed results.

Reviewer 2 Report

The work presents a framework for the assessment of robustness of machine learning algorithms. The general framework and its methodological basis are clearly presented.

Results obtained by applying the approach to a use case are carefully commented and discussed.

Overall the manuscript is scientifically sound and interesting.

I woul suggest detailing more on the rationale behind the choice of the different type of noise. How the noise types have been chosen? Why did the authors consider only Gaussian and replacement noise? Did they try to simulate the effect of different type of noise?

These information can be helpful to guide the reader through the interpretation of the specific aim of the study and its results.

Reviewer 3 Report

In this article, the authors applied ML classifiers with regard to their robustness by focusing on the variability of the classifier performance and changes in the classifiers’ parameter values using factor analysis and Monte Carlo simulations.

The paper is not well organized and well-proof-read, making it hard to understand. The presentation needs considerable improvement.

This submission raises several concerns, and many improvements-corrections are required.

1)In the introduction, the AI/ML acronyms are not defined.

2)At the end of the introduction, the authors should note the structure of the article.

3)The dataset is not analyzed. A feature analysis of the dataset needs to be done. Is this dataset balanced?

4) The authors focused only on LDA, SVM, RF, PLS-DA, and LR models. The evaluation should also be extended by applying AdaBoostM1, Rotation Forest, Multilayer Perception, Random Tree, Naive Bayes models and Ensemble methods such as Bagging, Voting and Stacking.

5)There is no comparison with related works based on the same dataset in order to provide a validation of the experimental results. A comparative analysis based on features, datasets, or algorithms is necessary.

6)The Appendix of this article should be in the main body of the article as contains important information about the parameters and a literature review of the ML models.

7)The way the discussion is written does not give specific information about the accuracy of the models. At the end, which model is superior?

8) For discussion, I would like to know the limitations and potential issues of this study. The academic implications of this study are not reported.

9)The conclusion section it is not presented any numeric information related to accuracy and other advantages of the proposed solution.

10)The authors miss the experiment setup. Please, demonstrate the environment of the experiment in detail.

11)Extensive editing of English language and style required.

12)There are too many references which are not updated 

Round 2

Reviewer 3 Report

I have no additional remarks on the revised version.

The authors have addressed my concerns.